# Beneficial Shifts in Gut Microbiota by *Lacticaseibacillus rhamnosus* R0011 and *Lactobacillus helveticus* R0052 in Alcoholic Hepatitis

**DOI:** 10.3390/microorganisms10071474

**Published:** 2022-07-21

**Authors:** Haripriya Gupta, Sung Hun Kim, Seul Ki Kim, Sang Hak Han, Hak Cheol Kwon, Ki Tae Suk

**Affiliations:** 1Institute for Liver and Digestive Diseases, Hallym University, Chuncheon 24252, Korea; phr.haripriya13@gmail.com (H.G.); wow8885@naver.com (S.K.K.); 2Korea Institute of Science and Technology, Gangneung 25451, Korea; marineamigo@hotmail.com; 3Department of Pathology, Hallym University College of Medicine, Chuncheon 24252, Korea; drhsh74@hallym.or.kr

**Keywords:** alcoholic hepatitis, probiotics, lipopolysaccharides, microbiota

## Abstract

Gut microbiota performs indispensable functions in the pathophysiology of alcoholic hepatitis (AH). We investigated the effects of *Lacticaseibacillus rhamnosus* R0011 and *Lactobacillus helveticus* for gut microbial restoration toward eubiosis in patients with AH. A multicenter, double-blind, and randomized trial was conducted. Probiotics (n = 44) and placebo (n = 45) groups received, during 7 days, *L. rhamnosus* R0011/*L. helveticus* R0052 at 120 mg/day and placebo. All patients were hospitalized to ensure abstinence. Liver function, lipopolysaccharide level, and stool analysis were evaluated in patients before and after 7 days of treatment. At baseline, the dominant bacteria were Gram-negative in both groups which decreased after the probiotics treatment and exhibited a significant reduction in lipopolysaccharide level (*p* < 0.001). The probiotics ameliorated the Child–Pugh scores (*p* < 0.001). Furthermore, the probiotics group showed a decline in the levels of alanine aminotransferase and gamma-glutamyltranspeptidase (*p* < 0.05). The probiotics changed the gut microbial composition at various taxonomical levels. The proportion of *Bacteroidetes* (147%) was increased after 7 days of probiotics supplementation while *Proteobacteria* (30%) and *Fusobacteria* (0%) were decreased. Administration of *L. rhamnosus* R0011 and *L. helveticus* R0052 conceivably associated with restoration of gut microbiome in AH patients and improved AH by modulating the gut–liver axis.

## 1. Introduction

Alcoholic liver disease (ALD) is one of the causes of high mortality worldwide and is generated by excessive and long-term alcohol drinking [1]. According to the report of the WHO, 3 million deaths/year result from consistent and chronic alcohol drinking, which account for 5.3% of the total mortality in the world. Some patients develop alcoholic hepatitis (AH) when they consume alcohol continuously. Severe AH is related with about 30% mortality rate in a month [2]. Alcohol use disorder, once diagnosed, is socially neglected and associated with disability in the life period. Approximately, about 15% of deaths in the young age group (20–30) are related to alcohol consumption [3,4].

The gut microbiome is composed of a lot of bacterial strains and microorganisms that inhabit the stomach and intestine and maintain eubiosis in the human body [5]. The gut microbiome plays key roles in the human body such as for homeostasis, metabolism, circulation, immunology, and hormone regulation [6,7]. The liver is directly connected to the intestine by the portal vein and receives all nutrients directly from the intestine. Thus, substances including endotoxin, alcohol, or bacterial products (pathogen-associated molecular patterns) from the intestine can be harmful to the liver [8,9]. Dysbiosis leads to alterations in the gut microbiome that trigger gut disruption and translocation of harmful metabolites and microbiome itself. This event causes the progression of chronic liver damage including ALD [10,11]. Additionally, in ALD patients and animal models of ALD, elevated plasma endotoxin and increased intestinal permeability and increased production of inflammatory cytokines are typical findings [12,13].

Recent reports have demonstrated that the gut microbiome is associated with the occurrence, prevention, treatment, and prognosis of ALD. Increases of favorable microbiota by selectively checking their characteristics and activity could be an initial approach to establish an amiable microbial community in the host [14]. Probiotics are related with strains that are good to the human, controlling gut homeostasis [15]. The gut microbiome and the immune system can be modulated by probiotics and *Lactobacillus* and *Bifidobacterium* are frequently used strains [16]. In light of this, *Lacticaseibacillus rhamnosus* and *L Lactobacillus helveticus* have been exemplified in animal model studies [17,18]. However, the pathophysiological importance of this association has yet to be fully determined. Moreover, limited human data are available on AH because of difficulty in clinical trials. Therefore, to elucidate the beneficial effects of *lactobacilli,* this clinical study investigated the therapeutic effects of probiotics *lactobacilli* in patients with AH.

## 2. Materials and Methods

### 2.1. Study Design and Participants

Between December 2012 and January 2015, a randomized and clinical trial (NCT02335632) was performed at 4 centers to evaluate the effect of *L. rhamnosus* R0011 and *L. helveticus* R0052. Patients who were >20 years old, showed aspartate aminotransferase (AST)/alanine aminotransferase (ALT) > 1 and AST (ALT) level > 50 IU/L, and were alcohol-drinking (more than 40 g/day (women) and 60 g/day (men) during 1 week prior to screening were enrolled. Patients’ last drinking was within 2 days of enrollment. Since AH patients usually reveal poor compliance in taking clinical trials and performing management in an outpatient clinic, all patients were hospitalized for the study and were not permitted to consume alcohol for 1 week of admission. Patients who had viral hepatitis (A, B, or C), autoimmune disease, pancreatitis, delirium tremens, hemochromatosis, Wilson’s disease, drug-induced liver injury, tumors, antibiotics or drug use during the study, severe AH (modified discriminant function ≥ 32), or obesity (BMI > 30 kg/m^2^) were excluded. All patients were supplied basic nutritional support (protein, 1.2–1.5 g/kg/day and 35–40 kcal/kg/day) with conventional AH treatment.

A total of 100 patients were consecutively enrolled. The primary aim was determined by liver function tests after 1 week (Figure 1A). The secondary aims were determined by changes in lipopolysaccharide (LPS) and pro-inflammatory cytokines, stool culture, and stool polymerase chain reaction denaturing gradient gel electrophoresis. The study protocols conformed to ethical guidelines (1975 Declaration of Helsinki) and received the approval by the institutional review boards for human research of all participating centers. Informed consent for participation in the study was obtained from each patient. 

Baseline examinations such as family/alcohol history, ultrasonography or contrast-enhanced computed tomography, X ray, electrocardiography, routine blood tests (complete blood count, electrolytes, liver function, and viral marker), and endoscopy were performed. Blood analysis was performed using standard methodologies. Biochemical tests included bilirubin, ALT, AST, gamma glutamyl transpeptidase (γGT), alkaline phosphatase (ALP), albumin, sodium, total bilirubin (TB), prothrombin time, total protein, glucose, international normalized ratio, and total cholesterol. The Child – Pugh score, pro-inflammatory cytokines (tumor necrosis factor (TNF-α) and interleukin (IL)-10), and LPS were also checked at day 1 and day 7.

### 2.2. Study Treatment

During this study, patients were managed with routine alcohol detoxification treatment, including fluid therapy, controlled diet, and thiamine supplementation. All patients were treated with legalon (silymarin, 210 mg/day), Bukwang Pharm Co. Ltd., Seoul, South Korea. Enrolled patients who fulfilled the criteria were randomly assigned to receive 7 days of cultured *L. rhamnosus* R0011 and *L. helveticus* R0052 at 120 mg/day (Lacidophil) or placebo. Lacidophil and placebos of the same shape and size were manufactured at Pharmbio Korea Co., Ltd. Patients were blinded to the randomization. Pro-, pre-, or anti-biotics were not given during the admission period. Probiotics and placebo were administered 3 times a day for 7 days (Figure 1A).

### 2.3. Randomization and Blinding

Online randomization programs were used to generate the randomized numbers with permuted blocks of random sizes used for the study. Probiotics and placebo packs were labeled with numbers and provided to sites in identical methods. Research nurses assigned study numbers to patients in the order they were enrolled. All members were blinded to treatment assignment.

### 2.4. Quantitative Analysis of Cytokines and LPS

For the evaluations of inflammatory cytokines, homogenates of blood were processed with ELISA kits (Human TNF-alpha Quantikine ELISA Kit (cat NO DTA00C, R&D systems, Inc., Minneapolis, MN, USA) and Human IL-10 Quantikine ELISA Kit (cat NO D1000B, R&D systems, Inc., Minneapolis, MN, USA)). For the LPS measuring, the LPS/LOS ELISA Kit (cat NO E1526Ge, USCN Life Science, Inc., Austin, TX, USA) was utilized. All measurements were performed according to the manufacturer’s instructions. 

### 2.5. Sample Preparation and DNA Extraction

Human stools were stored in a −20 °C refrigerator as soon as the stool was received (1–2 g) using stool paper and stool box and transferred to a −80 °C refrigerator in 1 week. Stool samples (1 g) were suspended in 10 volumes of Buffer ASL (Qiagen, Hilden, Germany) and homogenized by using a TissueLyser Ⅱ (Qiagen) with 5 mm stainless steel beads. Genomic DNA from stools was extracted by utilizing QIAamp DNA Stool Mini Kit (Qiagen). All measurements were performed according to the manufacturer’s instructions. The concentration and purity of each extracted DNA were determined using a NanoDrop spectrometer ND-1000 (NanoDrop Technologies, Wilmington, DE, USA) and stored at −20 °C until further processing.

### 2.6. Polymerase Chain Reaction-Denaturing Gradient Gel Electrophoresis

The microbial community of the stools was evaluated by polymerase chain reaction-denaturing gradient gel electrophoresis (PCR-DGGE) of bacterial *16S* rRNA genes. Nested PCR amplification was performed to amplify *16S* rRNA fragments suitable for DGGE analysis. At the first PCR set, *16S* rRNA genes were amplified using a universal bacteria primer (pair 27F and 1492R). In the second PCR set, routine primers (GC341F and 518R) were utilized to augment the V3 region of the *16S* rRNA gene for the DGGE analysis. The DGGE was analyzed by a Dcode universal mutation detection system (Bio-Rad, Hercules, CA, USA). The PCR products (500 ng) were loaded onto 8% (*w*/*v*) polyacrylamide gels with a denaturing gradient ranging from 40% to 60% (100% denaturant defined as 7 M urea plus 40% [*v*/*v*] formamide). The gels were electrophoresed at 60 V for 18 h in 1 X TAE (40 mM Tris–acetate, 1 mM EDTA, pH 8.3) and kept at a constant temperature of 60 °C. After electrophoresis, the gels were stained with ethidium bromide and photographed with UV transillumination. The DGGE profiles were analyzed with BioNumerics software (Version 7.1, Applied Maths, St-Martens-Latem, Belgium).

The visible DGGE bands were extracted by a sterile surgical blade and analyzed by using a QIAquick gel extraction kit (Qiagen). These bands were re-amplified with primers 341F (without GC-clamp) and 518. The resultant PCR products were purified using a QIAquick PCR purification kit (Qiagen) and sequenced with an ABI PRISMTM BigDyeTM terminator V3.1 kit on an ABI 3730XL DNA sequencer (Applied Biosystems, Foster City, CA, USA). The sequences obtained from each band were edited and assembled using a BioEdit program. For performing similarity search with sequence data available from the DNA Data Bank of Japan (DDBJ, http://www.ddbj.nig.ac.jp, (accessed on 1 July 2014)), the final edited sequences were used with the BLAST program.

### 2.7. Stool Analysis for the Metagenomics

Stool DNA was extracted by using a QIAamp stool kit (cat. 51504) and amplification of the V3–V4 region of the bacterial *16S* rRNA gene was performed by using barcoded universal primers. PCR was done according to the following methods: a first denaturation at 95 °C/5 min, 20 cycles of 95 °C/30 s, 55 °C/30 s, and 72 °C/30 s, followed by a last extension at 72 °C/10 min. Purification of the amplicons was performed with an Agencourt AMPure XP system (Beckman, USA) and quantification of the purified amplicons was performed by utilizing PicoGreen and quantitative PCR. After pooling of the barcoded amplicons, sequencing was carried out using a MiSeq sequencer on the Illumina platform (ChunLab Inc., Seoul, Korea) according to the manufacturer’s specification.

Bacterial profiling was conducted with the *16S*-based Microbiome Taxonomy Profiling platform of EzBioCloud Apps (ChunLab Inc., Korea). After taxonomy profiling of samples, comparative analyzer of EzBioCloud Apps was utilized for the comparison analysis for the sample. Taxonomy assignment of the reads was conducted with ChunLab’s *16S* rRNA database (DB ver. PKSSU4.0) [19] OTU picking was conducted with UCLUST and CDHIT with 97% of similarity cutoff [20]. In the subgroup analysis, Good’s coverage, rarefaction, α diversity, and β diversity were analyzed by using a comparative MTP analyzer. All *16S* rRNA sequences were deposited in the ChunLab’s EzBioCloud Microbiome database and sequencing reads of the *16S* rRNA genes of this trial were deposited in the NCBI Short Read Archive under the bioproject number PRJNA532302. Profiling phylogenetic marker genes including the *16S* rRNA gene are a key tool for the microbial community study but do not provide direct evidence of a community’s functional capability. We used phylogenetic investigation of community by reconstruction of unobserved states, a computational approach to predict the functional composition of a metagenome using marker gene data and a database of reference genomes [21].

### 2.8. Statistical Analysis

This study was a clinical trial evaluating changes in liver enzymes (primary aim) and change of gut microbiome, LPS, pro-inflammatory cytokines, and stool culture (secondary aim) in patients with AH after probiotic administration. We assumed the sample size by calculating with the difference of 0.8 and standard deviation of 1.5 in the gut microbiome. An independent-samples *t*-test was used for comparison of continuous variables. Quantitative data were expressed as mean ± standard deviation unless otherwise stated. Inter-group comparisons were performed by independent samples t-test using GraphPad Prism version 8.0 for Windows (GraphPad Software, San Diego, CA, USA). Differences in group means were compared by paired *t*-test and ANCOVA. Data from the routine blood tests and stool cultures were analyzed with statistical software (SPSS, version 20.0, SPSS, Inc., Chicago, IL, USA). For all tests, *p* < 0.05 was considered significant.

In the taxonomic analysis, we further correlated taxa in stool of alcoholic cirrhosis (AC) patients by Kruskal–Wallis H test and the linear discriminant analysis (LDA) effect size (LEfSE) method, which is used to discover high-dimensional biomarkers among microbial communities. The LEfSe model pinpoints taxa which are differently abundant between groups and evaluates the effect size of each significantly different taxon. Using the LEfSe algorithm, taxonomic biomarker discovery of bacterial taxa that were differentially abundant in probiotics and placebo groups were first acknowledged and verified using the Kruskal–Wallis H test with adjustments for multiple comparisons (*p* < 0.05). The heatmap was generated by The heatmap was generated using GraphPad Prism version 8.0 for Windows (GraphPad Software, San Diego, CA, USA).

## 3. Results

### 3.1. Abstinence and Probiotics Improved Liver Function in Alcoholic Hepatitis

A total of 100 patients (probiotics: 50 and placebo: 50) were enrolled; among them, 11 patients (probiotics, 6 and placebo, 5) were excluded because of patient’s refusal (7), early discharge (3), and other reasons (1). Finally, a total of 89 patients were hospitalized and completed the study (Figure 1B). The mean age of the patients was 51 ± 9 years, and 83 (93%) were male (Figure 1). The baseline characteristics of patients are described in Table 1. Fifty-two (52%) patients were diagnosed with liver cirrhosis: 29 (56%) in the probiotics group and 23 (44%) in the placebo group. In the blood test, the mean levels of AST, ALT, and γGT were 159.2 ± 250.7, 101.6 ± 176.4, and 436.8 ± 466.0 IU/L, respectively at baseline. 

From our previous study, we demonstrated that 7 days of abstinence was a crucial therapeutic intervention for patients with AH. Interestingly, AST, ALT, γGT, and TB were significantly improved after the 7 days of administration in both groups (Table 2). All patients revealed improvement in liver enzymes by abstaining because all patients were hospitalized and not permitted to drink alcohol during the study. The ALP level and Child–Pugh score were significantly improved only in the probiotics group. The probiotics groups also showed a significant reduction (*p* < 0.01) in LPS levels on day 7. In particular, patients with cirrhosis showed a significant reduction in LPS levels on day 7 (*p* < 0.01) (Figure 2). There were changes in the levels of anti-inflammatory IL-10 and proinflammatory TNF-α after probiotics administration, but the changes were statistically insignificant (Appendix A).

### 3.2. Probiotics Alter the Gut Microbiota in Patients with Alcohol Hepatitis

In AH patients, changes in the microbiota were visualized by PCR-DGGE fingerprinting, which also demonstrated differences in the microbial colonies between the probiotics and placebo. One patient (patient A) who received placebo and probiotics at different times showed a similar pattern on day 0. On day 7, changes were seen in the placebo and probiotics group (Figure 3A). Placebo supplementation decreased the expression of *Bacteroides*
*fragilis*, *Phocaeicola vulgatus*, *Parabacteroides distasonis*, *Fusobacterium mortiferum*, and *Veillonella parvula.* In contrast, *Lactobacillus gasseri*, *Lachnospira eligens*, *Clostridiales* spp., and *Escherichia coli* were increased by placebo supplementation. The expression of *B. fragilis*, *F. mortiferum*, and *E. coli* was decreased by probiotics administration. *Faecalibacterium prausnitzii*, *Veillonella*
*dispar*, *V. parvula*, and *Enterobacter* spp. were increased after probiotics supplementation. 

In the placebo group, the number of thinned or missing bands was greater than the number of thickened or new bands, but the opposite finding was observed in the probiotics group (Figure 3B). In the summary, typical changes in the probiotics group included increased diversity and dominant strains. Regarding compliance with taking probiotics, we checked for presence of the ingested strains in the feces (Figure 3C), and PCR amplification of *L. rhamnosus* and *L. helveticus* was performed to detect the presence of the strains in the probiotics group.

The placebo group revealed relatively low similarity between day 1 and day 7 compared with the probiotics group (41 ± 19 vs. 67 ± 20) (Figure 3D). The dominant strains were *E. coli*, *Clostridiales* sp., *F. mortiferum*, *Veillonella* sp., *Salmonella enterica*, and *L. eligens* in the placebo group. *F. prausnitzii* and *F. mortiferum* were dominant strains in the probiotics group (Figure 3D).

### 3.3. Lactobacillus Supplementation Modulates Alcohol-Induced Dysbiosis

For *16S* rRNA analysis, we selected only cirrhosis patients (n = 5) because Child–Pugh scores was significantly improved in those patients. Therefore, for the characterization of the gut microbiota associated with AC, we compared the alpha diversity between the placebo-post (n = 3) and probiotics-post groups (n = 2) after 7 days of treatment. α diversity indices were negatively correlated and significantly declined in the placebo-post group compared with the normal control group (n = 3) (*p* < 0.05) after 7 days of treatment (Figure 4A). In addition, the cirrhosis control group (n = 5 (placebo-pre n = 3, probiotics-pre n = 2)) showed almost the same diversity as the placebo-post treated group. However, the species richness and diversity indices were improved in the probiotics-post treated group, but the changes were not significant.

The beta diversity analysis using the unweighted UniFrac distance revealed that the placebo-post-treated group was separated from that of the normal control group (*p* <0.01) (Figure 4B). The fecal microbiome structure and composition in the placebo-post group were significantly different from those of the probiotics treated and normal control groups. Colonization with *Firmicutes*, *Fusobacteria*, and *Proteobacteria* was significantly and negatively associated with more preserved liver function. At the phylum level, *Firmicutes* was the most abundant, contributing 52.3%, 67.4%, 35.7%, and 34.1% of the gut microbiota in the probiotics-post, placebo-post, cirrhosis control, and normal control groups respectively, followed by *Bacteroidetes* (41.2%, 12.0%, 34.1%, and 60.9%, respectively) and *Proteobacteria* (5.9%, 14.6%, 24.6%, and 4.3%, respectively) (Figure 4C). On the other hand, *Fusobacteria,* which was insignificant in the normal control group, was found to be increased by 3.7% and 4.8% (*p* < 0.05) in placebo-post and cirrhosis control groups, respectively, and after 7 days of treatment with probiotics, *Fusobacteria* was reduced to 0.02%, which was statistically insignificant. However, the overabundance of *Proteobacteria* and *Firmicutes* did not reach statistical significance in the placebo-post and cirrhosis control groups.

At the class level, placebo-post and cirrhosis control groups were enriched with *Fusobacteria_c* (3.7% and 4.8%, respectively), *Gammaproteobacteria* (13.5% and 23.3%, respectively), and *Clostridia* (44.2% and 25.5%, respectively), and a decrease in *Bacteroidia* (11.9% and 34.1%, respectively) was observed. By contrast, after treatment with probiotics, a subsequent reduction in the abundance of *Fusobacteria_c* (0.0%), *Gammaproteobacteria* (5.0%), and *Clostridia* (38.4%) was observed, while there was increase in the abundance of *Bacteroidia* (41.2%) (Figure 4D). In addition, the genus level also showed significantly different abundances in each group. The genera *Prevotella*, *Lachnospira*, *Oscillibacter*, *Alistipes*, *Faecalibacterium*, and *Eubacterium* were increased in the probiotics groups after 7 days of treatment, and *Enterobacter*, *Clostridium*, *Streptococcus and Fusobacterium* were significantly enriched in the placebo-post and cirrhosis control groups after 7 days (Figure 4E).

LEfSe detected two bacterial clades, *Bacteroidales* and *Clostridiales,* showing statistically significant and biologically consistent differences in the probiotics treated group in comparison with the placebo-post and cirrhosis control groups. In the placebo-post and cirrhosis control groups, among Gram-positive *Firmicutes,* the genus *Clostridium*_g21 and its respective species *Ruminococcus gnavus* were significantly more enriched and, conversely, they were significantly less abundant in the probiotics group after 7 days of treatment than in the normal controls. The LEfSe LDA score more informatively reordered these taxa relative to the *p* value found for these families, highlighting the *Bacteroidales* and *Clostridiales* clades within the orders *Bacteroidia* and *Clostridia,* respectively (Table 2). Additionally, LEfSe focused on the *Firmicutes* phylum, emphasizing other anaerobic genera within *Lachnospiraceae*. The species *Prevotella copri* from the genus *Prevotellaceae_uc,* order *Bacteroidales*; and the species *FM873843_s* from the genus *Eubacterium*_g7, family *Lachnospiraceae* were significantly enriched in the probiotics group compared with the placebo and cirrhosis control groups.

The differential enrichment of specific bacteria at the genus level was also analyzed by the Kruskal–Wallis H test in the probiotics group compared with the placebo group, which is shown in Table 3 and Table 4 and Figure 5. Probiotics supplementation increased Gram-positive bacteria (Figure 4G). The genera *Oscillibacter* (*Ruminococcaceae*), *Faecalibacterium* (*Ruminococcaceae*), *Eubacterium*_g7, *Eubacterium*_g20, *Eubacterium*_g21 (*Lachnospiraceae*), and *Alistipes* (*Rikenellaceae*) were increased in the probiotics group and were depleted or reduced in the placebo and cirrhosis control groups (Figure 4F).

### 3.4. Probiotics Reduced Endotoxin Levels by Improving the Composition of Microbiota

Probiotics supplementation increased *Alistipes*, *Eubacterium*, *Faecalibacterium*, *Oscillibacter*, *Porphyromonadaceae*, and *Prevotella* (Figure 5). As all patients were admitted and stopped drinking alcohol, their LPS levels were generally decreased. However, the level of LPS was significantly decreased in patients with AH with cirrhosis (Figure 2). The species *V. dispar*, *Megasphaera micronuciformis*, *V. parvula group*, *B. fragilis*, *Prevotella stercorea*, *P. distasonis*, *L. fermentum*, and *Roseburia inulinivorans* were increased by the probiotics supplementation and decreased by placebo (Table 4). Some strains showed opposite results in the taxonomic analysis. 

## 4. Discussion

In AH, persistent alcohol drinking trigger overgrowth of Gram-negative strains in the gut and consequently secrete endotoxin such as LPS, which damage the gut barrier and facilitate the translocation of endotoxin from the gut to the liver through the portal vein [22,23]. A previous report demonstrated that a reduction in the intestinal bacterial load and subsequent endotoxin exposure can also mitigate alcohol-related intestinal inflammation and neuroinflammation [24]. In our study, abstinence effectively reduced serum LPS levels, and this result was augmented by supplementation with *L. rhamnosus* R0011 and *L. helveticus* R0052. A mice study corroborated that probiotics supplementation with *L. rhamnosus* GG modulates intestinal barrier functions and subsequent bacterial translocation to the liver, consequently reducing the translocation of endotoxin and thus reducing hepatic inflammation and liver damage [25,26,27].

Our previous study demonstrated that 7 days of abstinence was critical therapeutic intervention for patients with AH [28], which was proven by other earlier studies [29,30]. Blood tests including AST, ALT, γGT, and TB revealed significant improvement after the 7 days of abstinence in the placebo group regardless of the probiotics therapy, indicating that abstinence is the most important therapeutic intervention for patients with AH. However, it cannot fully improve the liver damage caused by excessive alcohol drinking. In this study, probiotics supplementation increased the proportion of fecal *L. rhamnosus* and *L. helveticus,* which mitigated serum transaminase significantly after 7 days of treatment in comparison with the placebo group. Furthermore, treatment also improved liver enzymes, which suggests a potential short-term role in patients with AH and cirrhosis. Lata et al. suggested that probiotics may improve liver functions by restoring the intestinal microbiota of cirrhotic patients [31]. Here, 7 days of probiotics treatment significantly abated the Child–Pugh score in cirrhosis patients, which further suggests the potential importance of probiotics in the treatment of ALD.

Microbiota community profiling by PCR-DGGE fingerprinting has been widely used to examine both microbial communities and spatial variability in microbial diversity in the same type of microbial community. *Bacteroides* spp. such as *B.*
*fragilis*, *Phocaeicola coprocola*, and Gram-negative anaerobes generally contribute to human diseases [32,33]. Similar to our research results, supplementation with probiotics reduced the density of *Bacteroides* spp. in AH patients. In contrast to pathogenic strains, butyrate-producing genus-like groups belonging to *Clostridium* clusters, *F.*
*prausnitzii, L. eligens* and *Clostridiales* spp., have beneficial effects on the host [34,35]. In our results, butyrate-producing bacterial density was restored with *L. rhamnosus* R0011 and *L. helveticus* R0052. 

In particular, *Enterobacter* spp. and *E. coli* densities were also reduced in the probiotics group. This means that the presence of such bacteria from the *Enterobacteriaceae* family could be correlated with the prognostic markers for disease severity. *L. rhamnosus* attenuated *E. coli*-induced inflammation by reducing proinflammatory receptor expression [36], and *L. helveticus* modified enterohemorrhagic *E. coli* in the intestinal microenvironment, thus preventing the virulence effect caused by *E. coli* [37]. Taken together, the negative correlation of pathogenic bacteria with that of probiotics may provide opportunistic insights for developing probiotics-based therapies for ALD.

The taxonomic abundance of *Bacteroidetes* was significantly lower, and the proportion of *Proteobacteria* and *Fusobacteria* was higher in patients with cirrhosis [38]. Puri et al. provided insight into the microbiome signature in AH patients [39]. Because the human gut microbiome is dominated by *Bacteroidetes*, its decline is probably due to an increase in *Proteobacteria* and *Fusobacteria*, which were clearly significantly reduced after probiotics treatment in this study.

Enrichment of *Fusobacteria* in the stool was positively correlated with alcohol-induced cirrhosis [39,40], and this enrichment was diminished after 7 days of probiotics treatment in the current study. In addition, bacteria class including *Streptococcaceae, Lactobacillaceae*, *Clostridiaceae*, and *Fusobacteriaceae* were increased in the placebo and cirrhosis control groups. The decreased presence of beneficial populations, such as *Lachnospiraceae* and *Ruminococcaceae,* affected the clinical phenotype and prognosis in AH and cirrhotic patients. The genus *Prevotella* and family *Prevotellaceae* were associated with disease progression in the alcoholic cirrhosis [38,41]. *L. rhamnosus* GG supplementation modified the intestinal microbiome, causing an increase in the abundance of *Prevotella*, *Lactococcus*, and *Ruminococcus* and a decrease in *Escherichia* in children with antibiotic-associated gastrointestinal symptoms [42]. From the clinical studies, it was observed that the *Subdoligranulum* genus was reduced in cirrhotic patients compared with healthy controls [43], and this genus was found to be negatively correlated with glycated hemoglobin and positively correlated with high-density lipoprotein cholesterol [44]. After 7 days of probiotics, our results revealed a significant increase in the relative abundance of *Subdoligranulum*. From this clinical study, we identified key operational taxonomical units whose population changes were significantly relevant to cirrhosis and treatment and may serve as potential biomarkers in future studies.

This study has limitation that we do not have clear mechanistic insights into the interplay of specific pathogens responsible for dysbiosis and restoration of the gut microbiota which needs to be explored and thoroughly researched. Additionally, this study was very short term, only 7 days; however, no adverse effects were seen throughout this study. Therefore. a long-term trial must be conducted to assess the therapeutic efficacy and adverse effects in ALD patients. Future studies will be indispensable to characterize the functions and pathways of the gut microbiome in ALD.

## 5. Conclusions

This study exemplified a comprehensive analysis of the fecal microbiome composition in patients with AH and cirrhosis. We conclude that the fecal microbial abundance of the cirrhosis control groups was akin to that of the placebo group but was distinct from that of the probiotics group. This designates an hourglass signature of disease severity in the gut microbiome and signals an initial decrease in phylogenetic diversity accompanied by a moderate stage of the disease leading to systematic distribution in severely affected individuals such as those with cirrhosis and hepatocellular carcinoma. Moreover, through this study, we tried to assist in the development of therapeutics to aid in the reestablishment of the commensal microbiota in AH and alcohol-related cirrhosis patients. Lastly, this study might reveal a promising avenue in the field of probiotics for therapeutic interventions in patients with AH and cirrhosis.

## Figures and Tables

**Figure 1 microorganisms-10-01474-f001:**
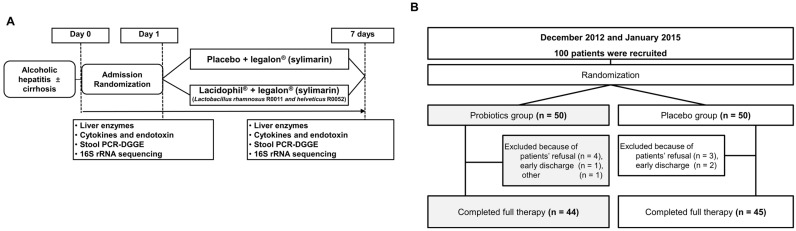
Study design and flow chart. (**A**) Schematic flow chart design of the study. (**B**) Randomization and allotment of the patients.

**Figure 2 microorganisms-10-01474-f002:**
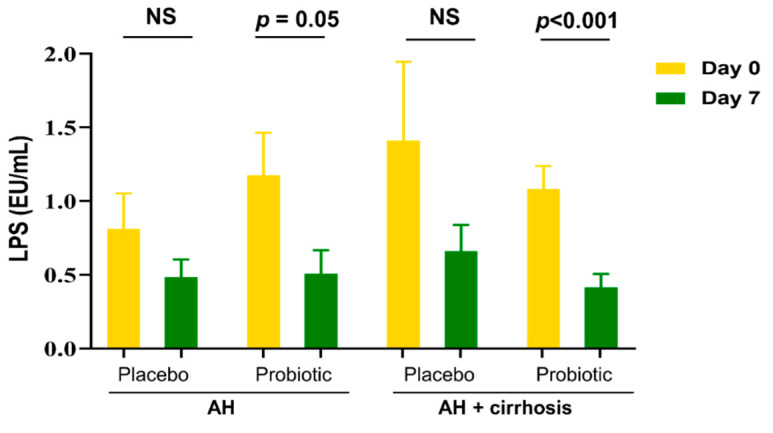
LPS level measurement in placebo and probiotics groups at day 0 and at day 7. LPS level in all patients (Lt) and cirrhosis and non-cirrhosis patients (Rt). Data are presented as mean ± standard error of the mean (SEM). *p* < 0.05 was considered significant. NS, non-significant; LPS, lipopolysaccharide.

**Figure 3 microorganisms-10-01474-f003:**
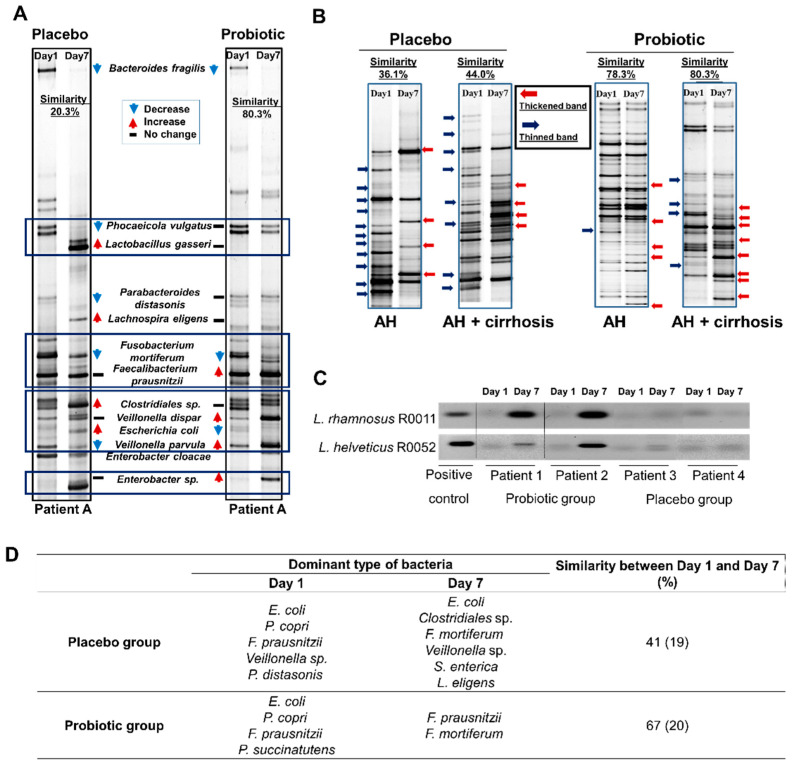
Dominant strains. (**A**) Electrophoresis result represents the microbial changes in patient A that was treated by both placebo (Patient 10) and probiotics (Patient 17) at different time points. (**B**) Change of diversity in both groups. Representative PCR-DGGE image used for analysis (**C**) PCR amplification for detection of *Lactobacillus*. (**D**) Dominant strains and similarity in placebo and probiotic group at day 0 and day 7. PCR-DGGE, polymerase chain reaction–denaturing gradient gel electrophoresis.

**Figure 4 microorganisms-10-01474-f004:**
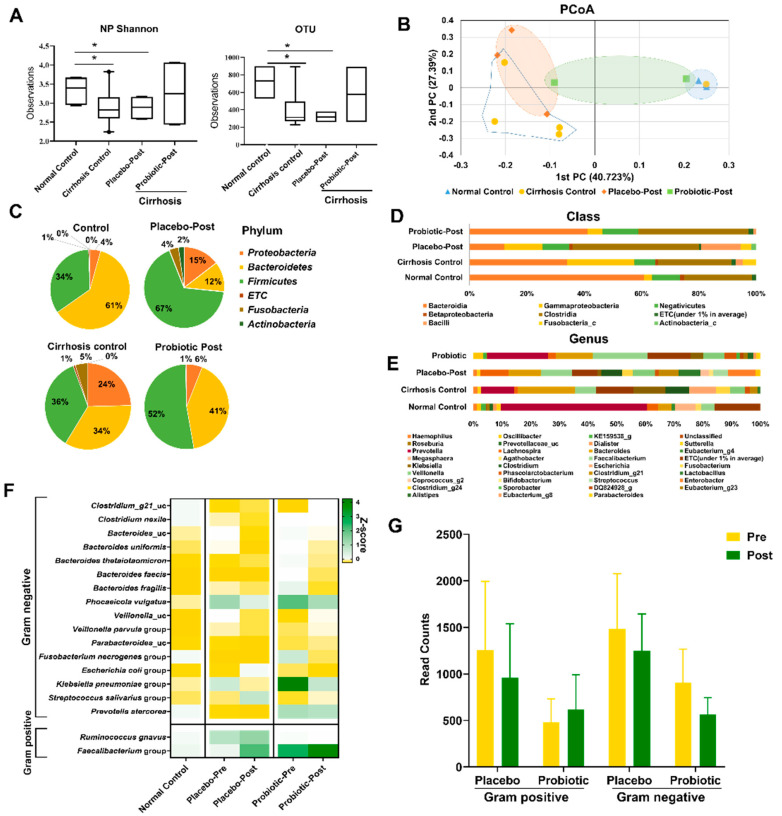
Taxonomic profiles. (**A**) Alpha diversity. Boxplot for alpha diversity representing the mean species diversity in placebo and probiotic treated groups compared with normal and cirrhotic control. (**B**) Principal coordinate analysis plot for beta diversity generated from generalized unifrac for genus. Microbial composition of placebo and probiotic treated groups compared with normal control and cirrhosis control at (**C**) phylum level, (**D**) class level, and (**E**) genus level. (**F**) Heatmap showing changes in species level before and after treatment. (**G**) Change of strains according to Gram’s classification. Data are presented as mean ± standard error of the mean. * represents significant value, *p* < 0.05. PC, principal coordinate; PCoA, principal coordinate analysis; OTU, operational taxonomic unit.

**Figure 5 microorganisms-10-01474-f005:**
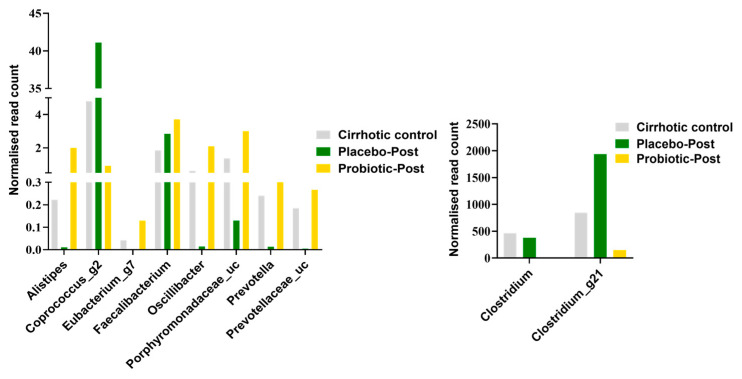
Microbial composition and LPS level measurement in placebo and probiotics groups at day 0 and at day 7. Comparative representation of selective genera between the groups, normalized by normal control group.

**Table 1 microorganisms-10-01474-t001:** Patients’ clinical baseline characteristics.

Variables	Values
Male (n (%))	83 (93)
Age (years)	50.8 ± 9.4
BMI (kg/m^2^)	22.9 ± 3.7
Presence of LC (n (%))	62 (52)
Total protein (g/dL)	6.7 ± 0.8
Albumin (g/dL)	3.8 ± 0.7
AST (IU/l)	159.2 ± 250.7
ALT (IU/l)	101.6 ± 176.4
ALP (IU/l)	118.0 ± 53.7
γGT (IU/l)	436.8 ± 466.0
TB (mg/dL)	2.3 ± 3.3
Cholesterol (mg/dL)	158.8 ± 46.1
PT (s)	12.8 ± 4.2
CP score in patients with LC	7.8 ± 2.5

Data are presented as mean ± standard error of the mean (SEM). n, number; AST, aspartate aminotransferase; ALT, alanine aminotransferase; ALP, alkaline phosphatase.

**Table 2 microorganisms-10-01474-t002:** Patients’ clinical characteristics and changes in liver function tests.

Variable (Mean ± SD)	Placebo (n = 45)					Probiotic (n = 45)				
	Pre		Post		*p* Value	Pre		Post		*p* Value
AST (IU/L)	126.2	±144.3	70.4	±73.3 *	0.010	192.3	±322.3	64.0	±48.5 *	0.008
ALT (IU/L)	91.3	±160.3	40.2	±40.5 *	0.010	111.9	±192.2	39.9	±37.2 *	0.004
γ-GT (IU/L)	434.7	±379.8	332.9	±314.5 *	0.002	439.0	±542.7	323.4	±399.7 *	0.008
Na (mEq/L)	139.0	±3.1	139.5	±4.2	0.480	138.3	±3.3	133.7	±29.9	0.330
TB (mg/dL)	2.5	±3.9	1.6	±3.0 *	<0.001	2.1	±2.6	1.4	±2.0 *	<0.001
Chol (mg/dL)	159.2	±46.1	168.8	±47.7	0.115	158.3	±46.6	164.8	±40.6	0.207
ALP (IU/L)	121.3	±55.0	113.4	±53.6	0.101	114.8	±52.8	104.0	±43.4 *	0.011
Albumin (g/dL)	3.8	±0.7	3.8	±0.6	0.615	3.7	±0.7	3.8	±0.6	0.258
TP (g/dL)	6.7	±0.8	6.7	±0.7	0.939	6.7	±0.8	6.8	±0.7	0.229
Glucose (mg/dL)	147.5	±79.1	136.2	±63.3	0.272	145.6	±69.4	144.8	±71.8	0.933
INR	1.2	±0.3	1.2	±0.3	0.737	1.2	±0.4	1.1	±0.4	0.146
PT (s)	12.8	±4.0	12.5	±3.3	0.447	12.9	±4.5	12.4	±3.9	0.239
Child-Pugh score ^a^	7.7	±2.5	7.2	±2.0	0.053	7.9	±2.5	7.2	±2.1 *	<0.001

* represents significant value, *p* < 0.05. Data are presented as mean ± standard error of the mean (SEM). ALT, alanine aminotransferase; AST, aspartate aminotransferase; γ-GT, gamma glutamyl transferase; TB, total bilirubin; Na, sodium; Chol, cholesterol; ALP, alanine phosphatase; TP, total protein; INR, international normalized ratio; PT, prothrombin time.

**Table 3 microorganisms-10-01474-t003:** Changes in taxonomic relative abundance according to LDA score among the groups.

Taxon Name	Taxon Rank	LDA	*p*-Value	Normal Control	Cirrhosis Control	Placebo-Post	Probiotic-Post
*Clostridium_g21*	Genus	3.520	0.023	0.004	2.568	6.983	0.666
*Ruminococcus gnavus*	Species	3.505	0.029	0.003	2.517	6.817	0.644
*Eubacterium_g7*	Genus	2.036	0.037	0.558	0.022	0.000	0.080
*FM873843_s*	Species	2.010	0.037	0.520	0.020	0.000	0.079
*Prevotellaceae_uc*	Genus	2.895	0.044	1.187	0.211	0.004	0.367
*Prevotella copri*	Species	2.010	0.045	0.167	0.021	0.000	0.054

LDA, linear discriminant analysis.

**Table 4 microorganisms-10-01474-t004:** Changes in relative abundance of genera in normal control, probiotic, and placebo groups.

Taxon Name	*p*-Value	Normal Control	Probiotic	Placebo	Change	
					Placebo	Probiotic
*Prevotella*	0.009	1.187	0.367	0.004	↓	↑
*Oscillibacter*	0.016	1.490	2.713	0.042	↓	↑
*Prevotellaceae_uc*	0.009	1.186	0.432	0.011	↓	↑
*Alistipes*	0.010	0.952	1.244	0.008	↓	↑
*Eubacterium_g7*	0.014	0.558	0.066	0.000	↓	↑
*Sporobacter*	0.035	0.409	0.852	0.010	↓	↑
*Subdoligranulum*	0.045	0.330	0.273	0.003	↓	↑
*Bacteroidaceae_uc*	0.049	0.316	0.258	0.105	↓	↑
*Parabacteroides*	0.047	0.240	1.265	0.333	↓	↑
*Ruminococcaceae_uc*	0.018	0.146	0.254	0.055	↓	↑
*Porphyromonadaceae_uc*	0.007	0.028	0.079	0.007	↓	↑
*Eisenbergiella*	0.024	0.200	0.181	0.020	↓	↑
*Clostridium_g21*	0.010	0.004	0.655	5.416	↑	↓
*Lactobacillus*	0.023	0.072	0.039	4.644	↑	↓
*Fusobacterium*	0.035	0.001	2.039	4.508	↑	↓
*Clostridium*	0.023	0.021	0.051	10.535	↑	↓
*Clostridiaceae_uc*	0.035	0.001	0.008	0.062	↑	↓
*Enterococcaceae_uc*	0.048	0.000	0.005	0.006	↑	↓
*Eubacterium_g20*	0.043	0.076	0.004	0.005	↑	↓

Abbreviations: ↑ indicates an increase in relative abundance in groups; ↓ indicates a decrease in relative abundance in groups.

## Data Availability

Not applicable.

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
