# Peer review of "Beneficial Shifts in Gut Microbiota by Lacticaseibacillus rhamnosus R0011 and Lactobacillus helveticus R0052 in Alcoholic Hepatitis"

_microorganisms, 2022, doi:10.3390/microorganisms10071474_

Round 1
Reviewer 1 Report
Authors evaluated that effects of administration of probiotic lactic acid bacteria on alcoholic hepatitis (AH), and on gut microbiota in AH patients.
This study is meaningful from the view points of curing AH without drugs. However, for publish, it is better to revise the manuscript more. Please refer following comments.
<Abstract>
・When authors show the result of statistics meaning p value, it is better to unify the style. Here, authors noted "p<0.001" in L20, and there are no half spaces between "P and <", and "< and 0.001". On the other hand, in L190, half spaces are inserted in "p < 0.05". Please check this point in entire manuscript.
・Authors showed the data in gut microbiota and monitored the change in both groups. For example, authors described that the proportion of Bacteroidetes (147%) was increased in probiotic group. However, these values are shown as relative ratio (%) in microbiota. How about the density of viable cells in gut in each group? Even if the percentage is higher, in some cases, the bacterial density is lower. Doesn't we need to consider bacterial density to evaluate probiotic effect? Is it OK to just focus on relative abundance (%)?
L99-105
L. rhamnosus and L. helveticus were cultured, and patients received these cultured cells at 120 mg/ day. In that case, 120 mg is dry weight? If it is, it is better to describe how to prepare dried cells.
Table 1
I recommend to modify the form of Table 1. For example, the position of "±" are different in each column. It is better to unify by adjusting the size of fonts and/or expand the length of each colum.
Figure 1
In Fig 1A, which DGGE picture does show placebo or probiotic? Please clearly indicate this as same with Fig 1B. And also, please correctly modify the position of "A" on the bottom. It is better to make it be located on right side after "Patient".
<References>
Please check the style of all references, and unify. For example, the title of journal in some references are abbreviated, but not in other references. And also, the scientific name is not noted as italics in some references.
Author Response
microorganisms-1814878
“Beneficial shifts in gut microbiota by Lacticaseibacillus rhamnosus R0011 and Lactobacillus helveticus R0052 in alcoholic hepatitis”
Reviewer 1:
Comment 1: Authors evaluated that effects of administration of probiotic lactic acid bacteria on alcoholic hepatitis (AH), and on gut microbiota in AH patients.
This study is meaningful from the view points of curing AH without drugs. However, for publish, it is better to revise the manuscript more. Please refer following comments.
Reply: On behalf of my team, I convey my best gratitude to Reviewer 1 for his/her comments, which helped us to improvise this manuscript relativity to the research and readability for better understanding of our work and findings in the scientific community.
Comment 2: <Abstract>
When authors show the result of statistics meaning p value, it is better to unify the style. Here, authors noted "p<0.001" in L20, and there are no half spaces between "P and <", and "< and 0.001". On the other hand, in L190, half spaces are inserted in "p < 0.05". Please check this point in entire manuscript.
Reply: We are very thankful to the reviewer for this comment pointing out variability in the manuscript. We have updated and unified the style of statistical notes throughout the manuscript.
Comment 3: Authors showed the data in gut microbiota and monitored the change in both groups. For example, authors described that the proportion of Bacteroidetes (147%) was increased in probiotic group. However, these values are shown as relative ratio (%) in microbiota. How about the density of viable cells in gut in each group? Even if the percentage is higher, in some cases, the bacterial density is lower. Doesn't we need to consider bacterial density to evaluate probiotic effect? Is it OK to just focus on relative abundance (%)?
Reply: We are grateful to the reviewer for this valuable and reasonable comment. We did metagenomic analysis of stool samples obtained from patients. For that DNA was isolated from the stool samples which was sequenced to obtain bacterial signatures. These sequences were further analyzed to obtain relative abundance of bacterial species in each patient’s stool samples. So, getting viable bacterial cells from humans and measuring density was not feasible. Also, results in relative abundance are widely accepted in the scientific community so we pursued and analyzed the results in relative abundance. Thus, we are requesting the reviewer to consider this argument against the comment. We hope we clarified the objective behind this comment.
Comment 4: L99-105
- rhamnosus and L. helveticus were cultured, and patients received these cultured cells at 120 mg/ day. In that case, 120 mg is dry weight? If it is, it is better to describe how to prepare dried cells.
Reply: We are very thankful to the reviewer for this important comment. We have used a company manufactured dry weight, already packed mix powder of Lacticaseibacillus rhamnosus R0011 and Lactobacillus helveticus R0052 under brand named Lacidophil. Since it was company manufactured and not cultured and freeze dried in our laboratory, we did not know the whole process behind the final product Lacidophil. Hence, we did not mention it in the method section. Thus, we are requesting the reviewer to consider this argument against the comment. We hope we explained the reason for this comment.
Comment 5: Table 1
I recommend to modify the form of Table 1. For example, the position of "±" are different in each column. It is better to unify by adjusting the size of fonts and/or expand the length of each column.
Reply: We are very thankful to the reviewer for this valuable suggestion. We have updated and adjusted the font size and symbols in Table 1 in the manuscript.
Comment 6: Figure 1
In Fig 1A, which DGGE picture does show placebo or probiotic? Please clearly indicate this as same with Fig 1B. And also, please correctly modify the position of "A" on the bottom. It is better to make it be located on right side after "Patient".
Reply: We are very thankful to the reviewer for this valuable suggestion. We have updated and modified the figures in the manuscript in Figure 1A and Figure 1B.
Comment 7: <References>
Please check the style of all references, and unify. For example, the title of journal in some references are abbreviated, but not in other references. And also, the scientific name is not noted as italics in some references.
Reply: We are grateful to the reviewer for this valuable comment. We have thoroughly checked again and updated the references as per the MDPI author guidelines in the manuscript.

Reviewer 2 Report
The manuscript 1814878 entitled “Beneficial shifts in gut microbiota by Lacticaseibacillus rhamnosus R0011 and Lactobacillus helveticus R0052 in alcoholic hepatitis” reports the impact of the administration of a combination of two lactic acid bacteria as probiotics to improve the health effects of alcoholic hepatitis. The authors appropriately described the approach used, and appointed the limitations of the current study. The results sound scientific and the manuscript is recommended for publication. Following are the recommendations for improvement.
Line 15-16 change “groups received 7 days of…” to “groups received during 7 days ..”
Line 18 and all over the manuscript change gram to Gram (this is a family name)
Line 38 the microbiome is the collective genomic content of a microbiota which in turn is a microbial community that inhabits a specific environment (consisting of bacteria, fungi, protozoa, viruses). The term microbiome also indicates the genetic capacity of a community. Please rewrite according.
Line 60 and 61 lactobacilli – not in italic
Line 81- tools of a computerized method- more details are required
Line 108 Program (computer)- what programme? see also the comment above.
Line 132 16S rRNA- italic, modify all over the manuscript
Line 183 change An independent- samples to An independent-sample
Line 191-194- Reference is required.
Line 194-199- Reference is required.
Line 207-208 – check the sentence
Line 209- 126 ± 144, 91 ± 160, and 434 ± 379 IU/L - indicate these values with more precision in both groups.
Line 239 delete E. coli
Line 240 and all over the manuscript follow the rules in Microbiology that to mention the scientific names of microorganisms are: the first time mention the full name (genus and species) in italic with the genus with first letter capitalized, subsequently the name should be written with the genus (abbreviated) followed directly by the species, both in italic. Example Bacteroides fragilis, then B. fragilis
Line 279-280 Indicate the correlation value to suppor the sentence: Colonization with Firmicutes, Fusobacteria and Proteobacteria was significantly and negatively associated with more preserved liver function
Line 353-354 - this information must be mentioned in the MM section and also recall the reader in the result section.
Line 363 change microflora to microbiota
Line 369-370 about Bacteroides coprocola - Revise the taxonomy, consult García-López, M.; Meier-Kolthoff, J.P.; Tindall, B.J.; Gronow, S.;Woyke, T.; Kyrpides, N.C.; Hahnke, R.L.; Göker, M. Analysis of 1,000 Type-Strain Genomes Improves Taxonomic Classification of Bacteroidetes. Front. Microbiol. 2019, 10.
Line 377 change densities was to densities were
Line 394 it is this correct? Lactobacillaceae members are a pathogenic group? Moreover, in the context of your study?
Line 410 change gut flora to gut microbiota
Line 411 change to be explore to to be explored

Author Response
microorganisms-1814878
“Beneficial shifts in gut microbiota by Lacticaseibacillus rhamnosus R0011 and Lactobacillus helveticus R0052 in alcoholic hepatitis”
Reviewer 2:
Comment 1: The manuscript 1814878 entitled “Beneficial shifts in gut microbiota by Lacticaseibacillus rhamnosus R0011 and Lactobacillus helveticus R0052 in alcoholic hepatitis” reports the impact of the administration of a combination of two lactic acid bacteria as probiotics to improve the health effects of alcoholic hepatitis. The authors appropriately described the approach used, and appointed the limitations of the current study. The results sound scientific and the manuscript is recommended for publication. Following are the recommendations for improvement.
Reply: On behalf of my team, I convey my best gratitude to the Reviewer 2 for his/her comments, which helped us to improvise this manuscript relativity to the research and readability for better understanding of our work and findings in the scientific community.
Comment 2: Line 15-16 change “groups received 7 days of…” to “groups received during 7 days ..”
Line 18 and all over the manuscript change gram to Gram (this is a family name).
Reply: We are very thankful to the reviewer for this valuable suggestion. We have updated the manuscript as per suggestion.
Comment 3: Line 38 the microbiome is the collective genomic content of a microbiota which in turn is a microbial community that inhabits a specific environment (consisting of bacteria, fungi, protozoa, viruses). The term microbiome also indicates the genetic capacity of a community. Please rewrite according.
Reply: We are grateful to the reviewer for this valuable and reasonable comment. We agree with the reviewer that the microbiome comprises all of the genetic material within a microbial community which inhabits a specific environment. Hence, we have updated the manuscript thoroughly and changed the “microbiota” with “microbiome” as per the requirement of the manuscript.
Comment 4: Line 60 and 61 lactobacilli – not in italic
Reply: We are very thankful to the reviewer for this valuable suggestion. We have updated the manuscript as per suggestion.
Comment 5: Line 81- tools of a computerized method- more details are required
Line 108 Program (computer)- what programme? see also the comment above.
Reply: We are grateful to the reviewer for this reasonable comment. We have done randomization through online free randomization program. We inserted patient’s number and it provided the grouping of patients at that the beginning of trial. Thus, we are requesting the reviewer to consider this argument against the comment. We hope we explained the reason for this comment.
Comment 6: Line 132 16S rRNA- italic, modify all over the manuscript.
Line 183 change An independent- samples to An independent-sample
Reply: We are very thankful to the reviewer for this valuable suggestion. We have updated the manuscript as per suggestion.
Comment 8: Line 191-194- Reference is required.
Line 194-199- Reference is required.
Reply: We are very thankful to the reviewer for this valuable suggestion. We have updated the manuscript as per suggestion and added the reference in the method section at Line 193-134.
“The LEfSe model pinpoints taxa which is differently abundant between groups and evaluates the effect size of each significantly different taxon [22].
“22. Segata, N.; Izard, J.; Waldron, L.; Gevers, D.; Miropolsky, L.; Garrett, W.S.; Huttenhower, C. Metagenomic biomarker discovery and explanation. Genome Biol 2011, 12, R60, doi:10.1186/gb-2011-12-6-r60.”
Comment 9: Line 207-208 – check the sentence
Reply: We are very thankful to the reviewer for this valuable suggestion. We have updated the manuscript as per suggestion and corrected at Line 206-208 and added Table 1 for the baseline characteristics of the patients.
Before correction:
“The base patients’ characteristics are described in Table 1. Sixty-two (52%) patients were diagnosed with liver cirrhosis: 29 (55%) in the probiotics group and 23 (44%) in the placebo group. In the blood test, the mean levels of AST, ALT, and γGT were 126 ± 144, 91 ± 160, and 434 ± 379 IU/L, respectively.”
After correction:
“The baseline characteristics of patients are described in Table 1. Fifty-two (52%) patients were diagnosed with liver cirrhosis: 29 (56%) in the probiotics group and 23 (44%) in the placebo group. In the blood test, the mean levels of AST, ALT, and γGT were 126 ± 144, 91 ± 160, and 434 ± 379 IU/L, respectively at baseline.”
Table 1. Patient’s clinical baseline characteristics
Comment 10: Line 209- 126 ± 144, 91 ± 160, and 434 ± 379 IU/L - indicate these values with more precision in both groups.
Reply: We are very grateful to the reviewer for highlighting the discrepancies in the manuscript. The above-mentioned comment line with value indicates the mean values of the patients at the baseline. We have updated the correct baseline values.
Before correction:
“In the blood test, the mean levels of AST, ALT, and γGT were 126 ± 144, 91 ± 160, and 434 ± 379 IU/L, respectively.”
After correction:
“In the blood test, the mean levels of AST, ALT, and γGT were 159.2 ± 250.7, 101.6 ± 176.4, and 436.8 ± 466.0 IU/L, respectively at baseline”
Comment 11: Line 239 delete E. coli
Line 240 and all over the manuscript follow the rules in Microbiology that to mention the scientific names of microorganisms are: the first time mention the full name (genus and species) in italic with the genus with first letter capitalized, subsequently the name should be written with the genus (abbreviated) followed directly by the species, both in italic. Example Bacteroides fragilis, then B. fragilis
Reply: We are very thankful to the reviewer for this valuable suggestion and reasonable comment. We have revised thoroughly the manuscript and updated the scientific names of bacterial species according to the Microbiology rules.
Comment 12: Line 279-280 Indicate the correlation value to support the sentence: Colonization with Firmicutes, Fusobacteria and Proteobacteria was significantly and negatively associated with more preserved liver function.
Reply: We are grateful to the reviewer for this reasonable comment. We have not done the correlational analysis. We have added this line as a reference to support our results that showed compositional reduction in phyla Firmicutes, Fusobacteria and Proteobacteria after 7 days of probiotic administrations. Reference article has been cited during manuscript revision. The cited article is mentioned below”
“24. Puri, P.; Liangpunsakul, S.; Christensen, J.E.; Shah, V.H.; Kamath, P.S.; Gores, G.J.; Walker, S.; Comerford, M.; Katz, B.; Borst, A., et al. The circulating microbiome signature and inferred functional metagenomics in alcoholic hepatitis. Hepatology 2018, 67, 1284-1302, doi:10.1002/hep.29623.”
Thus, we are requesting the reviewer to consider this argument against the comment. We hope we clarified the objective behind this comment.
Comment 13: Line 353-354 - this information must be mentioned in the MM section and also recall the reader in the result section.
Reply: : We are very thankful to the reviewer for this important suggestion. We have revised the and added reference to the result section so as to recall the results while reading in the discussion section as well. Below is the line added in the result section at Line 219-220.
“From our previous study, we have demonstrated that 7 days of abstinence was crucial therapeutic intervention for patients with AH [23].”
Comment 14: Line 363 change microflora to microbiota
Reply: We are very thankful to the reviewer for this suggestion. We have updated the manuscript as per suggestion.
Comment 15: Line 369-370 about Bacteroides coprocola - Revise the taxonomy, consult García-López, M.; Meier-Kolthoff, J.P.; Tindall, B.J.; Gronow, S.;Woyke, T.; Kyrpides, N.C.; Hahnke, R.L.; Göker, M. Analysis of 1,000 Type-Strain Genomes Improves Taxonomic Classification of Bacteroidetes. Front. Microbiol. 2019, 10.
Reply: We are grateful to the reviewer for this valuable and reasonable comment. We have updated the bacterial names with the most recent updated names. Following are the changed names of the bacterial species.
Before correction After correction
Bacteroides vulgatus Phocaeicola vulgatus
Eubacterium eligens Lachnospira eligens
These bacterial names have been updated throughout the manuscripts as well as in Figure 2, heat map in Figure 3 and supplementary figure 1C.
Comment 16: Line 377 change densities was to densities were
Reply: We are very thankful to the reviewer for this suggestion. We have updated the manuscript as per suggestion.
Comment 17: Line 394 it is this correct? Lactobacillaceae members are a pathogenic group? Moreover, in the context of your study?
Reply: We are very thankful to the reviewer for this valuable and reasonable comment. Lactobacillaceae family doesn’t have known pathogenic mechanism, nevertheless some bacterial species from Lactobacillaceae family has been reported to be pathogenic causing endocarditis, bacteremia, and pleuropneumonia [Reference: Members of the Lactobacillus Genus Complex (LGC) as Opportunistic Pathogens: A Review Microorganisms. 2019 May 10;7(5):126. doi: 10.3390/microorganisms7050126.]. However, we have corrected the sentence in the discussion section from “pathogenic bacteria” to “bacterial class” in Line 402 since there is no reported paper to support pathogenicity of Lactobacillaceae in alcoholic liver disease. Thus, we are requesting the reviewer to consider this argument against the comment. We hope we have provided justification for changes on behalf of the reviewer's comment.
Comment 18: Line 410 change gut flora to gut microbiota
Line 411 change to be explore to to be explored
Reply: We are very thankful to the reviewer for this suggestion. We have updated the manuscript as per suggestion.
